# Explainable AI for Estimating Pathogenicity of Genetic Variants Using Large-Scale Knowledge Graphs

**DOI:** 10.3390/cancers15041118

**Published:** 2023-02-09

**Authors:** Shuya Abe, Shinichiro Tago, Kazuaki Yokoyama, Miho Ogawa, Tomomi Takei, Seiya Imoto, Masaru Fuji

**Affiliations:** 1Artificial Intelligence Laboratory, Fujitsu Research, Fujitsu Ltd., Kawasaki 211-8588, Japan; 2Department of Cell Processing and Transfusion, IMSUT Hospital, The Institute of Medical Science, The University of Tokyo, Tokyo 108-8639, Japan; 3Division of Health Medical Intelligence, Human Genome Center, The Institute of Medical Science, The University of Tokyo, Tokyo 108-8639, Japan

**Keywords:** explainable AI, deep learning, knowledge graph, precision medicine, cancer genomic medicine

## Abstract

**Simple Summary:**

To treat diseases caused by genetic mutations, such as mutations in genes and cancer cells, genomic medicine is being promoted to identify disease-causing variants in individual patients using comprehensive genetic analysis (next-generation sequencing, or NGS) for diagnosis and treatment. However, clinical interpretation of the large amount of variant data output by NGS is a time-consuming task and has become a bottleneck in the promotion of genomic medicine. Although AI development to support this task has been conducted in various fields, none has yet been realized that has both high estimation accuracy and explainability at the same time. Therefore, we propose an AI with high estimation accuracy and explanatory power, which will eliminate the bottlenecks in genomic medicine.

**Abstract:**

Background: To treat diseases caused by genetic variants, it is necessary to identify disease-causing variants in patients. However, since there are a large number of disease-causing variants, the application of AI is required. We propose AI to solve this problem and report the results of its application in identifying disease-causing variants. Methods: To assist physicians in their task of identifying disease-causing variants, we propose an explainable AI (XAI) that combines high estimation accuracy with explainability using a knowledge graph. We integrated databases for genomic medicine and constructed a large knowledge graph that was used to achieve the XAI. Results: We compared our XAI with random forests and decision trees. Conclusion: We propose an XAI that uses knowledge graphs for explanation. The proposed method achieves high estimation performance and explainability. This will support the promotion of genomic medicine.

## 1. Introduction

Genetic diseases are caused by genetic variants transmitted from parents to children, whereas cancer is a disease resulting from the accumulation of variants. For the diagnosis and treatment of diseases caused by these variants, genomic medicine has been promoted to identify disease-causing variants in individual patients through comprehensive genetic analysis (next-generation sequencing, or NGS) [1,2].

The translation and clinical interpretation of a large amount of variant data output by NGS, by referring to medical papers, databases, and clinical trial information, is a very labor-intensive task and has become a bottleneck in the promotion of genomic medicine.

The application of AI in the medical field has produced significant results, including the prediction of lymph node metastasis in breast cancer and the progression of Alzheimer′s disease [3,4]. From these, AI is expected to eliminate bottlenecks in genomic medicine as well.

However, AI enables the identification of disease-causing genetic variants, but even if this is achieved with high accuracy, there is always the possibility that the AI-based estimation results may be incorrect [5,6,7,8,9,10]. Therefore, trusting the AI estimation results may lead to a misdiagnosis. Therefore, it is necessary for physicians to verify the accuracy of AI estimation results before using them for diagnosis [8]. Not only in the medical field, but for this application of verifying AI estimation results, a technology has been proposed to allow AI users to verify the estimation results by outputting the reasons that led to the estimation results along with the AI estimation results. This technology is called explainable AI (XAI) and has been intensively studied in recent years [11,12,13,14,15,16].

There are two challenges in achieving gene variant identification with XAI. The first is the difficulty of combining high estimation accuracy with high explainability, which is a general problem for explainable AI. In terms of explainability for XAI, AI can be classified into black-box AI and white-box AI: estimation AIs with high performance but low explanatory power are black-box AIs, while AIs with low estimation performance but high explanatory power are white-box AIs. AI development to support this work is being conducted in various fields, but nothing has been achieved that has both high estimation accuracy and explainability at the same time [17]. The second challenge is that it is unclear what explanations physicians should provide for the use of AI in diagnosis. For example, one method of explanation in XAI is to output the internal information of the AI. However, while such an explanation may be convenient for AI developers, it is incomprehensible to physicians who are unfamiliar with AI. Physicians require an explanation that they can understand using their own knowledge.

We aimed to address these issues by adding an explanatory function to the black-box AI to achieve high explainability while maintaining high estimation performance. This additional functionality is achieved by combining the black-box AI estimation results and accompanying information with genomic medical knowledge. To this end, various databases on genomic medicine are maintained and integrated, and the information that explains the data itself (ontology) is assigned to these databases to construct a knowledge graph. With this knowledge graph, a large amount of knowledge about genomic medicine can be made available through computer programs, and in doing so, high estimation performance and explanatory power can be achieved (Figure 1). To solve the bottleneck of genomic medicine, we propose XAI, which combines high accuracy and explainability, and provides physicians with understandable explanations. We report the results of the application of our AI in the identification of disease-causing variants.

## 2. Materials and Methods

### 2.1. Knowledge Graph–Integrating Genome Databases

Physicians surveyed medical articles, public databases, and clinical trial information and guidelines such as ACMG [18], JSH [19], ASH [20], and NCCN [21] for clinical interpretation and translation. To achieve XAI using these databases in an integrated manner, these databases were combined, given an ontology, and a knowledge graph was constructed. In this study, a knowledge graph was constructed using public databases with relatively well-developed data.

To build our knowledge graph, we adopted the Resource Description Framework (RDF)-style [22] ontology defined by an open community. The open community is called med2rdf and is led by Kyoto University and the Database Center for Life Science. To integrate each database, a hub node for genes and variants was prepared, and the nodes of the genes and variants in each database were connected to the hub node. This allowed different databases to be used centrally, making large amounts of genomic medical data available from XAI and achieving high estimation performance and explanatory power.

The knowledge graph we constructed consisted of each database, an ontology for computer programs to understand the data, and hub nodes to connect various databases. The databases were ClinVar [23] (April 2020 version; 703,471 records; 337,480,558 triples), dbNSFP [24] (v4.1; 83,293,580 records; 10,588,087,867 triples), COSMIC [25] (V91; 36,741,586 records; 1,981,773,447 triples), and dbscSNV [24] (v1.1; 15,030,435 records; 431,501,716 triples; dbscSNV is included in dbNSFP), for a total of 13,338,843,588 triples. The ontologies used were Med2rdf [26] (142 triples), SIO [27] (15,608 triples), HCO [28] (42,140 triples), and Faldo [29] (235 triples), for a total of 58,125 triples. Our ontology refers to other basic ontologies commonly used in RDF, but we have not counted them as triples. The hub nodes connecting the databases were 2,224,371,765 triples. A total of 15,563,273,478 triple knowledge graphs were constructed. This is larger than the 103 million triples of DBpedia, a well-known large-scale knowledge graph [30].

### 2.2. Explainable AI

The proposed explainable AI is a method that adds an explanation mechanism to the black-box AI. This explanation mechanism combines the explanatory information of the black-box AI with the physician’s knowledge to provide an explanation that is in line with the physician′s usual investigation. To achieve this, we require an AI that is capable of handling graphical data, such as knowledge graphs. Therefore, deep learning, which learns and estimates from the graph data, was employed. Some of these deep learning methods output the degree to which the nodes on the graph contribute to the estimation (contribution) as an explanation of the estimation. We believe that it is inappropriate to use this contribution degree as is and to emphasize the nodes in the graph data with high contribution degrees to explain the estimation results. This is because such an explanation may: (1) lack information for the explanation (lack of explanatory information); (2) not consider the important points in the explanation (lack of consideration of importance); or (3) be a form of explanation that is unfamiliar to physicians.

Lack of explanatory information: Presenting only the highly contributing nodes is insufficient as an explanation, and there are often cases where physicians cannot understand the meaning of the nodes or why they are the reason for the estimation result unless related nodes are presented simultaneously. For example, if the contribution of a node in pathogenicity is high, presenting only this node does not improve comprehensibility for physicians. In this case, it is necessary to explain the name of the variant to which the pathogenic node is connected and the fact that this information is from the ClinVar database, thereby simultaneously presenting the related information.Lack of consideration of importance: It is important to present the information that physicians consider important. For example, if a variant is listed in ClinVar or COSMIC, the fact that it is listed provides important information to the physician.Unfamiliar format: Normally, physicians report the results of their literature searches and verify medical information in writing. Likewise, the literature and guidelines to which physicians refer are written, and physicians rarely have the opportunity to refer to graphical data. Therefore, graphical data are a form of expression unfamiliar to physicians, and explanations using graphical data may be difficult for physicians to understand. Explanations using text, a data format familiar to physicians, are important. Thus, our XAI generates text from graph data and presents explanations.

The following explanations were realized for the above issues:Explain the node with the highest contribution to deep learning.When explaining a node, the node necessary for understanding should be explained simultaneously.The information that physicians consider important should also be explained.Explain according to the guideline of ACMG.

To achieve these goals, the following information was defined for each type of node using an ontology:Importance of each node to the physician (X-Impact).A rule (X-Rule) consisting of a combination of the following information:A rule for the set of nodes required to describe any node.Rules for generating sentences from node sets.

We constructed a subgraph consisting of all nodes that were directly or indirectly referenced by the variant under study and applied all X rules to all node sets. The importance of the generated sentences (X-Factor) was calculated from the contribution (X-Contribution) and X-Impact during deep learning estimation using the following formula:X-Factor = sigmoid((X-Contribution × X-Impact))(1)

The generated sentences were sorted in order of increasing X-Factor, and the top N sentences were used as explanatory sentences.

We employed deep tensor as a deep learning implementation that could handle graph data [31]. Deep tensor outputs contributions to the edges during estimation (X-Contribution_edge_). Since our method assumes that nodes have contributions, we converted edge contributions to node contributions using the following equation:X-Contribution = X-Contribution_edge_(2)

Figure 2 shows an example of this calculation.

### 2.3. Evaluation Method

#### 2.3.1. Target of Evaluation

The performance of estimating whether a variant was the cause of the disease (pathogenicity) was evaluated. Various methods have been proposed for this estimation depending on the type of variant, among which single nucleotide missense variants are known to be difficult to estimate [32]. Therefore, we evaluated the estimation performance for this missense variant.

For evaluation, we estimate the pathogenicity of variants and measure their correctness. For this purpose, we prepared a set of variants with known pathogenicity as our correct answer set.

#### 2.3.2. Correct Answer Set

The correct answer set was generated in ClinVar, which assigns each variant a pathogenicity and a review status, expressed as stars 0–4. This review status represents the method used to determine the pathogenicity of the variant and can be considered a confidence indicator of the accuracy of the pathogenicity. The more stars, the higher the confidence level; if two or more institutions agree on the pathogenicity determination, the review status is two or more stars. Therefore, those with two or more stars were used.

We also excluded variants that were determined to be likely pathogenic, likely benign, or the variant of uncertain significance (VUS), and used only those variants that were determined to be pathogenic or benign. The total number of variants was 5568, including 2158 pathogenic variants and 3410 benign variants.

#### 2.3.3. Five-Fold Cross-Validation

To measure the performance of AI, we train on a set of correct answers in advance and measure estimation performance on a set of correct answers. However, if the same data is used for training and estimation, estimation performance for unknown data cannot be measured. Therefore, we trained on a portion of the set of correct answers and measured estimation performance on the remaining set of correct answers. Thus, the set of correct answers was divided into five parts, the estimation performance was measured for each partition, and the average of the five estimation performances was used as the performance of the method (five-fold cross-validation).

However, simply dividing the set of correct answers into five is problematic. Since variants have different properties for each gene, if we train on a variant of one gene and estimate on another variant of the same gene, we may not be able to correctly measure the performance of estimating an unknown variant. If trained on a variant of one gene, a variant of another gene should be estimated. Therefore, the set of correct answers was divided into five sets so that the training and estimation would target different chromosomes, although this would be a larger unit than genes.

#### 2.3.4. Balancing the Number of Pathogenic and Benign Variants

When learning, better results are expected if the number of pathogenic and benign variants is equal. Therefore, we randomly excluded benign variants from the set of correct answers and equaled the number of pathogenic and benign variants. The result was a final correct set of 2158 pathogenic variants and 2158 benign variants, for a total of 4316 variants.

#### 2.3.5. Trained Model

We prepared the correct answer for the estimation from ClinVar. Therefore, if information from the ClinVar is used for features during training, estimation performance may not be measured correctly. On the other hand, the information from the ClinVar is useful for explaining the estimation results.

Therefore, we decided to prepare a trained model to measure the estimation performance and a trained model for explanation. The trained model for estimation does not include information on ClinVar, while the trained model for explanation does.

Since the information in the ClinVar contains the correct answer, simply training with the information in the ClinVar may always train a model that explains only the information in the ClinVar. In this case, it will not be possible to explain variants that are not described in ClinVar. To avoid this, we constructed a learned model in which half of the randomly selected variants were explained using information from the ClinVar and the other half of the variants were explained without information from the ClinVar.

#### 2.3.6. Correct Answer Set for Explanation

For the evaluation of explainability, the correct answer was used, which was slightly different from the evaluation of estimated performance. Variants with ambiguous correct answers were excluded from the set of correct answers in order to correctly evaluate the performance in the estimation performance evaluation. However, since a larger amount of training data are better, variants that were likely pathogenic and benign were included in the set of correct answers in the trained model for explanation.

Furthermore, since there was no need to limit the type of variants to missense variants, we used data from all single nucleotide variants and used 30,754 pathogenic and 10,909 benign variants as the correct set. To reduce data bias, we also randomly excluded pathogenic variants and used 10,909 pathogenic and 10,909 benign variants as the correct set for explanation.

#### 2.3.7. Features for Estimation

The deep tensor used in the proposed method can directly handle graph structures. Therefore, we obtained the graph of features from our knowledge graph and performed training, estimation, and explanation.

For the variant to be estimated and its neighbors within 100 bases of this variant, we constructed a subgraph consisting of the following feature nodes. Then, between these subgraphs, links were added from the nodes of the variant to be estimated to the nodes of the neighboring variants. Thus, a graph of the features of the variants to be estimated was constructed.

In addition, some features of dbNSFP were not included in the trained model for estimation in order to avoid false improvements in estimation performance due to problems caused by circularity [33].

The features used in learning and estimating the learned model for estimation were as follows:COSMIC: registration status, sample size, number of papers;DbNSFP: scores for each algorithm;SIFT [34];LRT [35];PROVEAN [36];PhyloP100way_vertebrate [37];GERP++_RS [38];SiPhy_29way_logOdds [39].

In addition, the following information on genetic variants within 100 bases of the studied genetic variant was also used as a feature:ClinVar: clinical significance, review status, year of last update, number of submissions, publications, and sources of publications;COSMIC: registration status, number of samples, and publications.

#### 2.3.8. Features for Explanation

In addition to the features used in the learned model for estimation described above, the following features were also used to train and explain the training model for explanation:ClinVar: clinical significance, review status, year of last update, number of submissions, publications, and sources of publications;dbNSFP: scores for each algorithm;CADD [40];DANN [41];FATHMM [42];M-CAP [43];MetaLR [8];MetaSVM [8];MutPred [44];MutationTaster [45];Polyphen2_HDIV [46];Polyphen2_HVAR [46];REVEL [6];Fathmm-MKL_coding [42];PhastCons100way_vertebrate [47];DbscSNV: Scores of each algorithm;AdaBoost;Random forest.

#### 2.3.9. Methods of Comparison

The following methods were compared to evaluate estimation performance:Deep tensor: A black-box AI and deep learning method that uses graph data. We used this in our proposed method;Decision trees: A white-box AI said to have excellent explanatory properties;Random forests: A black-box AI with excellent estimation performance.

#### 2.3.10. Features for Decision Trees and Random Forests

While deep tensor can directly handle graph structures as features, decision trees and random forests cannot handle graph structures. Therefore, we converted the graph structure features, constructed by deep tensor, into a format that can be handled by decision trees and random forests.

To compare performance under equivalent conditions, it is desirable to use exactly the same features. However, in order to convert the data into a format that can be handled by decision trees and random forests while preserving the full information of the graph structure, it is necessary to consider all combinations of features. This would result in huge amounts of data containing a large number of useless feature combinations. Such features are impractical since they require a large amount of computation and are likely to significantly degrade estimation performance. Therefore, we analyzed the data in advance, conducted preliminary experiments, and determined the feature values based on these results. However, we used the same features as deep tensor whenever possible.

The format of features for decision trees and random forests is an array of pairs of feature names and their values, and the number of elements in the array does not change. Since the number of neighboring variants changes, the feature values for these variants were converted based on the knowledge obtained from prior experiments.

The three points for converting features are as follows: (1) features that change the number of elements in the array are aggregated, (2) the combination of features should be limited to useful ones, and (3) other features should be converted as much as possible from the graph information.

Specifically, the features of the variants to be estimated were prepared as an array of feature names and their value pairs. However, for example, the number of elements is indefinite when simply converted to an array, since the features for literature are different nodes with different literature information. Therefore, the number of references was used as the feature value.

However, since the literature included in ClinVar has publisher information, we tabulated the data by publisher, whereas the COSMIC literature does not have such information, so we simply totaled the data.

Since the number of neighboring variants was indefinite, the total was used as the feature, but since the proximity to the variant to be estimated is important, it was summed for each difference in distance from the variant to be estimated. The number of features was calculated for each distance between the variant and the target variant, since the proximity of the variant to the target variant is important.

The number of samples in COSMIC was carried forward by log_10_ of the number of samples, aggregated as follows:If number of samples = 1, aggregate to the number 0;If 1 < number of samples ≤ 10, aggregate to the number 1;If 10 < number of samples ≤ 100, aggregate to the number 2;If 100 < number of samples ≤ 1000, aggregate to the number 3;If 1000 < number of samples ≤ 10,000, aggregate to the number 4;If 10,000 < number of samples, aggregate to the number 5; All cases greater than 10,000 were aggregated to 5.

The following features of ClinVar were also aggregated:VariantImpact and clinicalsignificance;VariantImpact, reviewStatus, and clinicalsignificance.

#### 2.3.11. Other Experimental Conditions

The proposed method takes a variant as input, and outputs the degree of pathogenicity of the variant as a score from 0.0 to 1.0, where 0.5 or more is interpreted as pathogenic and less than 0.5 as benign.

An X-Rule was developed to generate explanatory text for the estimation results, with input from physicians (Table 1).

We present the explanatory sentences generated by our proposed method for the cases of hereditary diseases and cancer. We also present explanations by the decision trees, for comparison. Note that we do not provide an explanation for random forests, since it is an unexplainable method.

Figure 3 shows the pipeline of our method described so far.

## 3. Results

### 3.1. Evaluation of Estimation

We compared the estimation performance of deep tensor (version 20191210_mod_bugfix), decision trees, and random forests (decision trees and random forests using the scikit-learn [48] version 0.24.0 implementation). Estimation performance was 0.94 (93 epochs), 0.93 (27 depths), and 0.91 (9 depths) for deep tensor, random forests, and decision trees, respectively (Table 2, Figure 4 and Figure 5). Deep tensor showed higher estimation performance than decision trees and random forests.

### 3.2. Evaluation of Explicability

Using variant MYO7A NM_000260.4(MYO7A):c.5618G>A (p.Arg1873Gln) as an example, we compare the proposed method with the decision tree explanation. This variant causes hereditary hearing loss. It is also not described as pathogenic in the version of ClinVar used in our experiments, but is described as pathogenic in the recent 202005 version of ClinVar. Therefore, if we can correctly estimate the pathogenicity of this variant, it is an example of correctly estimating an unknown variant.

The decision trees correctly estimated this variant as pathogenic. Figure 6 shows the model of the decision tree training results visualized using PyDotPlus (version 2.0.2). The paths that explain this variant are circled with red boxes. This red box was drawn manually. Figure 7 shows the result of the human interpretation of the path of this variant in text. Our proposed method also correctly estimated this variant as pathogenic and generated the explanation shown in Figure 8.

We present variant SLC4A1 NM_000342.4(SLC4A1):c.216G>T (p.Glu72Asp). This is not registered in any of the databases we have used, such as ClinVar. This variant has been suspected as a cause of blood cancers, but has been clinically confirmed to be benign. With the proposed method, we were able to correctly infer that this variant is benign. Figure 9 is a description of the proposed method, showing that this variant is benign.

In addition, we also present variant TP53 NM_001126112:exon4:c.G250A:p.A84T. This variant is also not described as benign in the version of ClinVar we used in our experiments, but is described as benign in the recent 202005 version of ClinVar. And we were able to correctly estimate it as benign using the proposed method. Figure 10 is a description of the proposed method.

## 4. Discussion

### 4.1. Results of the Performance Evaluation

The proposed method achieved a higher estimation accuracy than random forests, which are generally known for their superior estimation performance. Simultaneously, the proposed method generated sentences similar to human interpretations of the tree structure of decision trees. Thus, we confirmed that the proposed method achieves the same, or better, explanatory power than decision trees.

As shown in Figure 4, the decision trees are highly accurate when the depth parameter is around 5, but less accurate when the depth parameter is greater than that. This is thought to be due to overlearning. On the other hand, deep tensor and the random forests do not change their accuracy significantly when the parameter is changed, suggesting that they are able to learn without overlearning. In addition, deep tensor consistently maintains higher performance in comparison to random forests, when the parameter is varied. This is thought to indicate that deep tensor achieves higher generalization performance without overlearning compared to random forests.

### 4.2. Comparison of Explanations

Figure 6 is the training result of the decision tree and is also the output of the decision tree explanation. Figure 7 is the human interpretation of this, taking into account the information of the relevant variant and expressed in sentences.

In a decision tree, each node is an element of the explanation. Each node then determines the next node based on whether it meets or does not meet a threshold value. Tracing back to the last node reveals the pathogenicity that is the result of the classification. Thus, when looking at each node in isolation, it is unclear how the explanation of each node affects the determination of pathogenicity. It is also unclear whether the threshold values for each node are appropriate for determining pathogenicity. Therefore, it does not make sense for a physician to independently verify the explanation of each node to validate the estimation results. It is necessary to verify that all nodes satisfy the threshold value at the same time. It is possible to verify whether several nodes are satisfied at the same time. However, in the case of Figure 6, it is necessary to verify whether six nodes are satisfied at the same time, which requires a lot of effort for a human to verify. In summary, in order to validate the decision tree explanation, it is necessary to verify that all sentences are satisfied simultaneously for each statement that does or does not clearly imply pathogenicity, which is difficult.

On the other hand, Figure 8 is an explanation of the proposed method. The second and third sentences of this explanation of the proposed method use information from dbNSFP to show that this variant is presumably pathogenic. dbNSFP is a database of scores from various algorithms that estimate the pathogenicity of a variant, with higher scores for each algorithm indicating a higher likelihood of pathogenicity. The fourth sentence suggests that this variant is not benign, based on the information that the variant of the same amino acid is VUS. The fifth and eighth sentences suggest that this variant may be pathogenic since the nearby variant is registered as pathogenic in ClinVar. Similarly, the seventh sentence also suggests that this variant may be pathogenic since the nearby variant is registered as pathogenic in COSMIC. As shown in Figure 9 and Figure 10, the proposed method can also generate explanations for various variants.

Thus, each sentence of the proposed method explains independently and with different evidence whether this variant is pathogenic. It is possible to verify the pathogenicity of this variation from this explanation since each statement can be verified independently. We believe that verifying the explanation of the proposed method is easier than verifying the explanation of the decision tree.

### 4.3. Estimation Performance of Other Methods

Various methods and benchmarks have been proposed for estimating the pathogenicity of mutations [3,4,5]. Among these, REVEL and ClinPred are well-known for their high estimation performance. Both methods are similar in that they are based on random forests, using dbNSFP as a feature. We would like to compare these methods with the proposed method, but it is not easy to compare machine learning methods [6,15]. The comparison results differ due to differences in training data, test data, and parameter tuning during training, but it is difficult to align these conditions. Fortunately, the results of ClinPred testing with ClinVar are publicly available, so it is possible to compare the proposed method with ClinPred. However, the version of ClinVar used by ClinPred for training and testing differs from the version of ClinVar used by us, as do the criteria for selecting the records in ClinVar. Therefore, the comparison is not made under the same conditions.

The AUC listed in the ClinPred paper is 0.98, and the AUC of the proposed method is also 0.98. From this point of view, the performance is equivalent.The mutations in the test results of ClinPred were matched with the mutations in the test results of the five-fold cross-validation of the proposed method, and the performance of ClinPred was calculated only from the mutations that could be matched. The mutations that could not be matched were 30 of 863, 8 of 863, 14 of 863, 8 of 863, and 6 of 864, respectively. Since only a few mutations could not be matched, we do not believe that they had a significant impact on the performance measures. The performance of ClinPred in these results was accuracy 0.96, precision 0.99, recall 0.93, F1 score 0.96, and AUC 0.99. This performance is higher than the performance of ClinPred described in their paper [15]. We believe that differences in training data are the cause of the differences in performance. ClinPred has excluded some mutations from the test to avoid false performance increases due to type 1 circularity [15,33]. On the other hand, the proposed method excludes such mutations from the training data for the same reason. From this, we believe that the increase in performance of ClinPred is due to the fact that it differs from the test set used in their paper [15].

Based on these results, we believe that the proposed method has an estimation performance close to that of ClinPred. In addition, these studies aim for high estimation performance; we aim for both high performance and explainability.

### 4.4. Realization of Explanations by Random Forests

The output of random forests does not explain the estimation results. However, there are possible ways to realize the explanation. For example, the deep tensor used in the proposed method uses LIME to output the edges of the graph that contributed to the estimation results, and the proposed method uses the information on these edges to provide an explanation [49]. Similarly, it is possible to apply LIME or SHAP to Random Forest and output the features that contributed to the estimation results [50]. Therefore, by using this information, it may be possible to achieve the same explanation in random forests as in the proposed method.

However, we have constructed a knowledge graph integrating various data in genomic medicine in order to provide physicians with understandable explanations. To utilize this knowledge graph, we believe that the proposed method using deep tensor, which can directly handle graph data, is more appropriate.

### 4.5. Another Knowledge Graph

We constructed a knowledge graph of genomic medicine to achieve an explanation. Other knowledge graphs for genomic medicine include CKG, which is a large knowledge graph integrating many databases in genomic medicine [51]. However, important databases on mutations, such as ClinVar, dbNSFP, and COSMIC, are not integrated. Also, the format of the graph is Neo4j, which is different from our knowledge graph based on standardized specifications. Despite these differences, both the CKG and ours are knowledge graphs that integrate various data for genomic medicine. In the future, we would like to combine both of them to construct a more valuable knowledge graph.

### 4.6. Application to Genome Medicine

The number of genetic mutations found by NGS can number in the tens of thousands in a single patient. The physician identifies a small number of pathogenic mutations among them to determine a course of treatment. In the process of identifying these mutations, physicians consult databases of various mutations. In addition to the mutation under investigation, they also investigate surrounding mutations and make a comprehensive judgment based on these results. Due to the large number of mutations to be investigated, this investigation takes a lot of time, and it is not practical to do it completely manually.

Therefore, only those mutations whose pathogenicity is clearly described in the database may be extracted mechanically, and only the extracted mutations may be judged by humans. For example, commercial reports such as FoundationOne CDx and Oncomine do not clearly disclose the method for determining pathogenicity, but given some public information, it is thought that pathogenicity is determined by a curated knowledge database and rules [52,53,54]. However, since only a few mutations are known to be pathogenic in the database, pathogenic mutations may be missed by this method.

To solve this miss, it is necessary to examine mutations whose pathogenicity has not been identified in the database. However, the number of such mutations is so large that manual investigation requires a great deal of time and is not realistic. Therefore, there are high hopes for an approach in which AI discovers mutations that are not listed in the database but are pathogenic, and physicians can determine only such mutations. However, in order to determine the pathogenicity of a mutation, it is necessary to investigate not only the mutation under investigation but also the surrounding mutations. Therefore, a lot of time is needed to determine the pathogenicity of a single mutation. Therefore, it is necessary not only to find pathogenic mutations, but also to explain the estimated results in order to support physicians in their investigations and judgments.

Our XAI will support such investigations and decisions by physicians. We believe that this will allow physicians to investigate and determine more pathogenic mutations and eliminate the bottleneck of investigating and determining mutations in genomic medicine.

However, the extent to which the proposed method can eliminate bottlenecks in genomic medicine needs to be evaluated in the future. For example, the degree of efficiency of the investigation, the number of mutations that can be investigated, and the number of patients that can be treated. However, evaluating these factors is difficult to implement since it would involve changes in the systems and workflows used by physicians. It also requires time for physicians to learn new systems. Another reason is that physicians involved in genomic medicine are extremely busy. Therefore, the proposed method needs to be implemented and evaluated in a practical system.

### 4.7. Application of AI in Future Genomic Medicine

To identify the pathogenicity of a mutation, physicians investigate various databases to make a decision. Therefore, the results of investigations and decisions may vary from physician to physician, as difficult decisions are made based on the results of many investigations.

For example, the ACMG has 16 criteria for classifying pathogenic variants, 12 criteria for classifying benign variants, and 15 rules for combining criteria [18]. Thus, a complex decision-making process is required.

In order for genomic medicine to be applied to more patients in the future, more physicians need to be involved in genomic medicine. At this time, it will be important to implement AI to homogenize the results of investigations and decisions.

### 4.8. Application to Other Fields

There are two challenges to applying the proposed method to fields, other than genomic medicine.

The first challenge is that the construction of X-Impact and X-Rule requires expertise in the applied field and labor-intensive manual work. In this application to genome medicine, there was no problem in terms of expertise since the project was carried out in collaboration with physicians. In addition, the manual construction of X-Impact and X-Rule in the application of genome medicine enables accurate explanations, which is expected to be an advantage in the medical field where accuracy is important.

The second challenge is that the construction of knowledge graphs requires know-how in selecting appropriate ontologies. As mentioned earlier, this was not a problem in this case since the project was conducted in collaboration with physicians.

To solve these issues, a method to automatically construct X-Impact and X-Rule should be considered. This would facilitate the application of X-Impact and X-Rule to new fields since it would not require expertise or manpower, and it would also enable the application of X-Impact and X-Rule to data that has not been maintained as a knowledge graph. However, in the case of automatic construction, ensuring the accuracy of explanations is an important issue.

## 5. Conclusions

In this study, we proposed XAI to solve the bottleneck of genomic medicine. The unique feature of this method is that it constructs a huge knowledge graph for genomic medicine and uses it to achieve estimation and explanation. We also proposed features that the description of XAI for genomic medicine should satisfy. The proposed method is then compared with decision trees and random forests to show that the proposed method combines the advantages of both white-box and black-box AI and can achieve high estimation accuracy and explainability simultaneously. The proposed method opens up the possibility of AI application to genomic medicine.

## Figures and Tables

**Figure 1 cancers-15-01118-f001:**
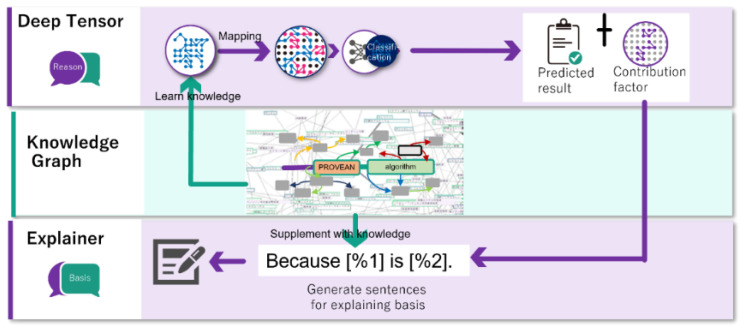
This is an overview of our explainable AI methodology using the knowledge graph and deep tensor.

**Figure 2 cancers-15-01118-f002:**
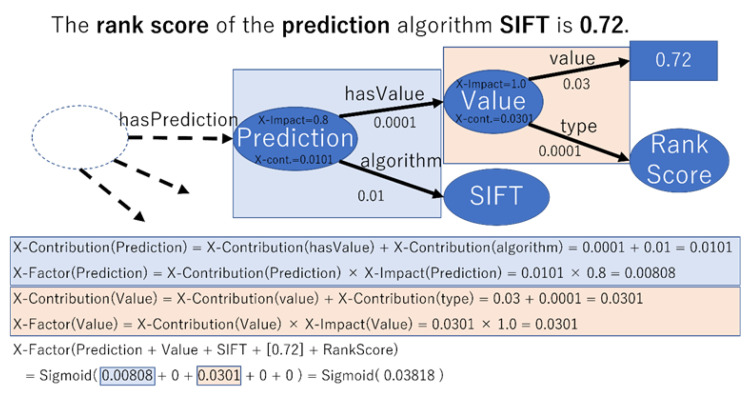
This illustrates the process of calculating X-Factor with an example.

**Figure 3 cancers-15-01118-f003:**
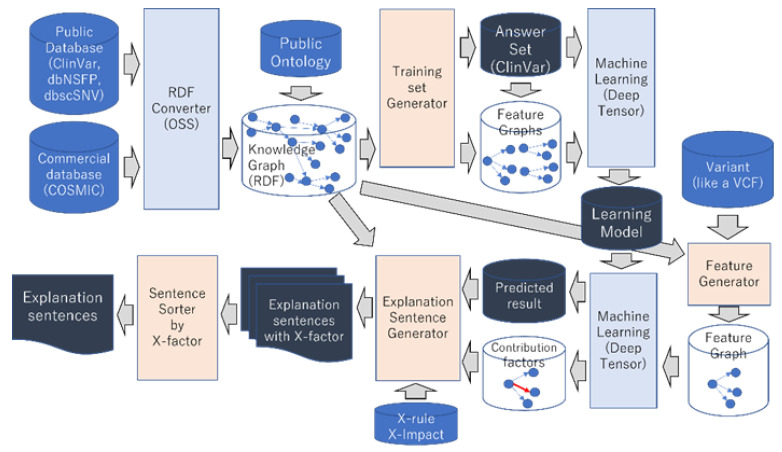
This shows the pipeline of our method.

**Figure 4 cancers-15-01118-f004:**
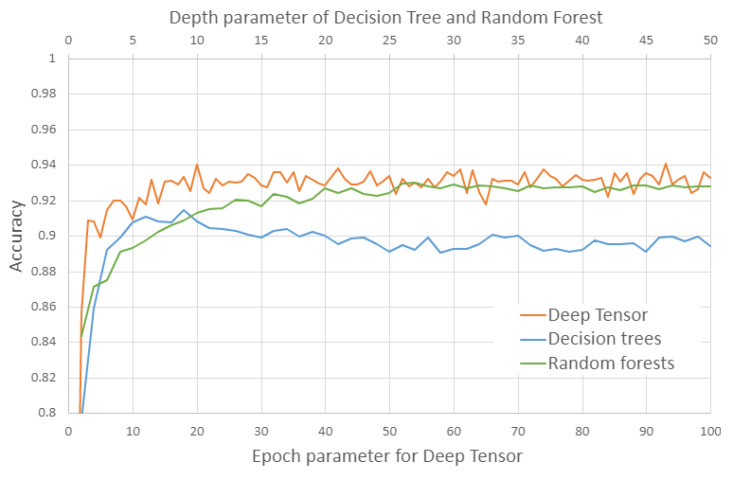
This details the estimation performance of deep tensor, decision trees, and random forests. The horizontal axis is the epoch parameter for deep tensor, and the depth parameter for decision trees and random forests. We varied the epoch or depth parameter and considered the highest estimation to be the performance of each method. These results are averages of a five-fold cross-validation.

**Figure 5 cancers-15-01118-f005:**
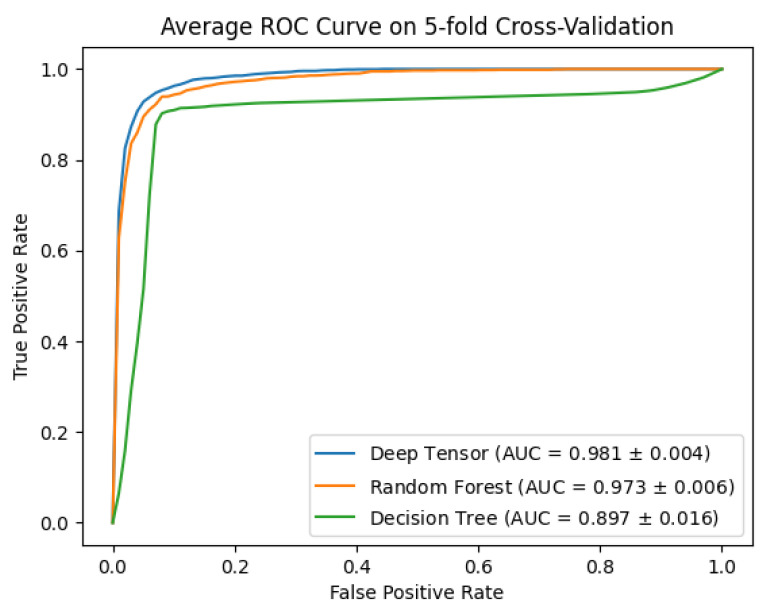
This shows ROCs for deep tensor, decision trees, and random forests.

**Figure 6 cancers-15-01118-f006:**
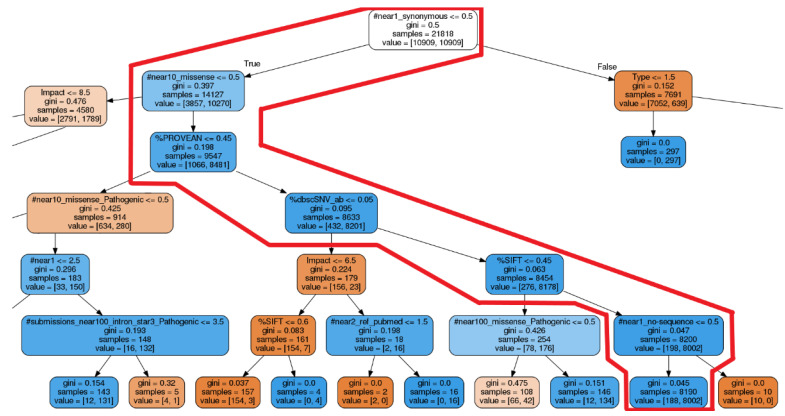
This is the decision trees model for the variant MYO7A NM_000260.4(MYO7A):c.5618G>A (p.Arg1873Gln) visualized using PyDotPlus (version 2.0.2). The paths circled in red are the subtrees that illustrate the case study. The red box was described manually. Several important nodes were enlarged for ease of viewing. This figure is an enlarged version of the red box area overall.

**Figure 7 cancers-15-01118-f007:**
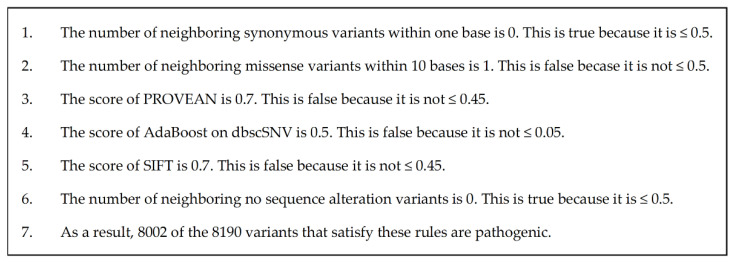
This figure is the result of human interpretation and writing of the path in the decision tree of MYO7A NM_000260.4(MYO7A):c.5618G>A (p.Arg1873Gln), circled in red in Figure 6, using information from the case study.

**Figure 8 cancers-15-01118-f008:**
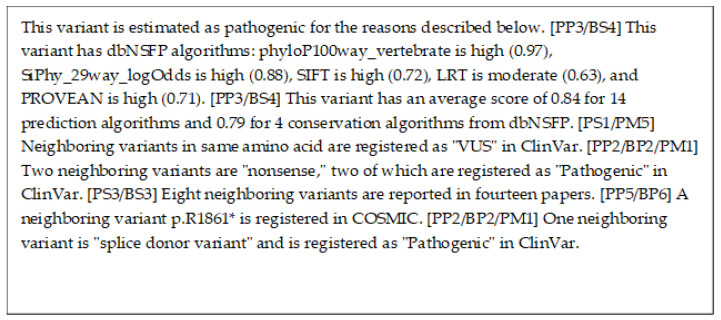
This is an explanation using our proposed method for the variant MYO7A NM_000260.4(MYO7A):c.5618G>A (p.Arg1873Gln). The "*" in "p.R1861*" refers to mutation to any amino acid.

**Figure 9 cancers-15-01118-f009:**
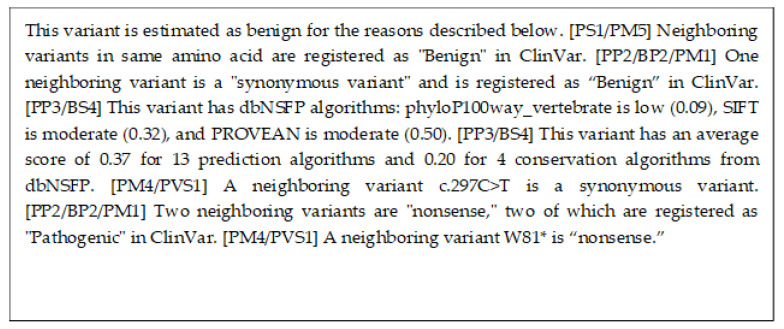
This is an explanation using our proposed method for the variant SLC4A1 NM_000342.4(SLC4A1):c.216G>T (p.Glu72Asp).

**Figure 10 cancers-15-01118-f010:**
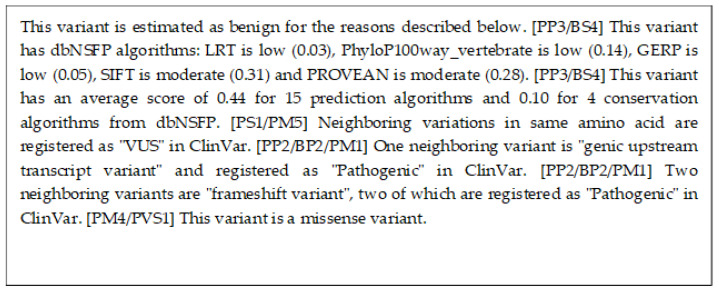
This is an explanation using our proposed method for the variant TP53 NM_001126112: exon4:c.G250A:p.A84T.

**Table 1 cancers-15-01118-t001:** This is the rule (X-Rule) for generating explanatory text with our proposed method. In "Variants in COSMIC", "*" in "p.R1861*" means mutation to any amino acid. The URL "https://pubmed.ncbi.nlm.nih.gov/28492532/" of "Variants reference" was confirmed on 9 February 2023.

X-Rule Name	Sample Sentence
Neighboring variants in ClinVar	Fourteen neighboring variants are “missense variant”, three of which are registered as “Pathogenic” in ClinVar.
Outstanding algorithms in dbNSFP	This variant has dbNSFP algorithms: phyloP100way_vertebrate is high (0.97), SiPhy_29way_logOdds is high (0.88), SIFT is high (0.72), PROVEAN is high (0.71), and LRT is moderate (0.63).
Average of algorithms in dbNSFP	This variant has an average score of 0.84 for 14 prediction algorithms, and 0.79 for 4 conservation algorithms from dbNSFP.
Same amino acid variants	Neighboring variants in the same base are registered as "VUS" in ClinVar.
Neighboring variant papers	Eight neighboring variants are reported in twenty-six papers.
Variants in COSMIC	A neighboring variant, p.R1861*, is registered in COSMIC.
Variants in dbNSFP	This variant has a score of 0.88 according to the SIFT algorithm of dbNSFP.
Variants reference	This variant is explained in https://pubmed.ncbi.nlm.nih.gov/28492532/ (accessed on 2 December 2022).
Variants in ClinVar	This variant is considered pathogenic with 3 stars in ClinVar, and the report has 3 submissions.
Variants type	This variant is a missense variant.

**Table 2 cancers-15-01118-t002:** This overview compares the performance of the proposed method with random forests and decision trees.

Method	Accuracy	Precision	Recall	F1 Score	AUC
Deep tensor	0.94	0.94	0.94	0.94	0.98
Random forests	0.93	0.93	0.92	0.92	0.97
Decision trees	0.91	0.92	0.90	0.91	0.90

## Data Availability

The data can be shared up on request. However, some data cannot be provided because this technology is to be commercialized.

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
