# Peer review of "Explainable AI for Estimating Pathogenicity of Genetic Variants Using Large-Scale Knowledge Graphs"

_cancers, 2023, doi:10.3390/cancers15041118_

Round 1

Reviewer 1 Report

The paper presents a KG-approach that is used for the prediction of genetic variants. As a feature, the results are explained in text form.

Major:

-       It is claimed that this approach will “eliminate the bottlenecks in genomic medicine”. This is neither shown nor realistic.

-       Do the explanations contribute to understanding how the method works? How the algorithm comes to the result is still a black box. I am not a genomic expert, but I was wondering if the outputs of the tool really help. Take Fig. 7 as an example -does this help a clinician?

-       Although cross-validation is mentioned, only a single value (accuracy) per model is given as a result. The authors should add the associated error across the folds.

-       Authors should provide sensitivity, recall, AUC.

-       The authors state that random forests are unexplainable (ll. 341-342). However, methods such as Shapley Additive Explanations (SHAP) exist to make random forests explainable. Please explain why this is not applicable.

-       The authors should compare their approach and the result with existing models such as ClinPred (PMID: 30220433).

-       The comparison between the proposed method and the baseline method is not fair: “Therefore, we used the 292 same features as Deep Tensor as much as possible; however, instead of converting them 293 to an array that perfectly preserved the graph structure, we processed some features based 294 on our knowledge so that decision trees and random forests could handle them easily. […] combined features were limited to those that were considered 298 useful”. Please consider a better baseline.

-       “In the evaluation of explainability, a slightly different set of correct answers was used 326 than in the evaluation of estimation performance. “ This seems adhoc, please justify.

-       „To estimate disease-causing variants using AI, various methods have been proposed 424 and compared using benchmarks under various conditions [3–5]” Please compare against these benchmarks.

-        

Minor:

-       Please do not refer to tables from the abstract and state quantitative results instead of “performance was high”.

-       Please perform spell checking / a native speaker check

-       Do the author know the clinical KG https://github.com/MannLabs/CKG ? How does it compare to the proposed approach?

-        Page 8, line 323: clinicalImpact is used twice

-       The methodology is hard to understand without code. Please consider adding it.

Author Response

Response to Reviewer 1 Comments

The authors would like to thank the reviewers and editors for their time and effort to review this manuscript.  Please consider the attached manuscript that has been revised according to the reviews.  Response to each reviewer's comments follows.

Major:

-       It is claimed that this approach will “eliminate the bottlenecks in genomic medicine”. This is neither shown nor realistic.

We have added a description of "eliminate the bottlenecks in genomic medicine" in section 4.4.

-       Do the explanations contribute to understanding how the method works? How the algorithm comes to the result is still a black box. I am not a genomic expert, but I was wondering if the outputs of the tool really help. Take Fig. 7 as an example -does this help a clinician?

We have added an explanation of how the output of our method can be useful in section 4.5.

-       Although cross-validation is mentioned, only a single value (accuracy) per model is given as a result. The authors should add the associated error across the folds.

-       Authors should provide sensitivity, recall, AUC.

We have added Table3 and Figure5.
We have added Precision, recall and AUC values. In addition, we added ROC.
Figure 4 describes the means of the five-fold cross-validation.

We have updated the scikit-learn version for these and re-run the experiment. The results have changed slightly but the trend is the same.

-       The authors state that random forests are unexplainable (ll. 341-342). However, methods such as Shapley Additive Explanations (SHAP) exist to make random forests explainable. Please explain why this is not applicable.

We have added a discussion of SHAP in section 4.3.

-       The authors should compare their approach and the result with existing models such as ClinPred (PMID: 30220433).

We have added a discussion of ClinPred in section 4.3.

-       The comparison between the proposed method and the baseline method is not fair: “Therefore, we used the 292 same features as Deep Tensor as much as possible; however, instead of converting them 293 to an array that perfectly preserved the graph structure, we processed some features based 294 on our knowledge so that decision trees and random forests could handle them easily. […] combined features were limited to those that were considered 298 useful”. Please consider a better baseline.

The description of the conversion from graph structures to arrays in section 2.5 we have changed so as not to be misleading.

-       “In the evaluation of explainability, a slightly different set of correct answers was used 326 than in the evaluation of estimation performance. “ This seems adhoc, please justify.

We have reviewed the description in section 2.3.

-       „To estimate disease-causing variants using AI, various methods have been proposed 424 and compared using benchmarks under various conditions [3–5]” Please compare against these benchmarks.

In section 4.2 we have added a comparison with ClinPred.

Minor:

-       Please do not refer to tables from the abstract and state quantitative results instead of “performance was high”.

We have corrected it.

-       Please perform spell checking / a native speaker check

We will be using an English editing service.

-       Do the author know the clinical KG https://github.com/MannLabs/CKG ? How does it compare to the proposed approach?

In section 4.3 we have added a discussion.

-        Page 8, line 323: clinicalImpact is used twice

We have corrected it.

-       The methodology is hard to understand without code. Please consider adding it.

We plan to apply this methodology to our product and offer it to hospitals. Therefore, it is difficult to publish the code.

Reviewer 2 Report

Title needs some modification. Currently, its somewhat confusing. 

In the introduction, what key theoretical perspectives and empirical findings in the main literature have already informed the problem formulation? What major, unaddressed puzzle, controversy, or paradox does this research address? 

What are some key issues that present study has addressed?

Authors should further clarify and elaborate novelty in their abstract.

Why your proposed model is important? What other similar model looks like from previous research? 

What are the practical implications of your research? 

What are the limitations of the present work?

Below papers have some interesting implications that you could discuss in your introduction and how it relates to your work.

Vulli, A.; et al.. Fine-Tuned DenseNet-169 for Breast Cancer Metastasis Prediction Using FastAI and 1-Cycle Policy. Sensors 2022, 22, 2988.

El-Sappagh, Shaker, et al. "Automatic detection of Alzheimer’s disease progression: An efficient information fusion approach with heterogeneous ensemble classifiers." Neurocomputing (2022).

Author Response

Response to Reviewer 2 Comments

The authors would like to thank the reviewers and editors for their time and effort to review this manuscript.  Please consider the attached manuscript that has been revised according to the reviews.  Response to each reviewer's comments follows.

Title needs some modification. Currently, its somewhat confusing.

In the introduction, what key theoretical perspectives and empirical findings in the main literature have already informed the problem formulation? What major, unaddressed puzzle, controversy, or paradox does this research address?

What are some key issues that present study has addressed?

We have added a description.

Authors should further clarify and elaborate novelty in their abstract.

We have reviewed the abstract description.

Why your proposed model is important? What other similar model looks like from previous research?

We have reviewed the description of the introduction.

What are the practical implications of your research?

In section 4.4 we have added an explanation.

What are the limitations of the present work?

In section 4.4 and 4.5, we have added an explanation.

Below papers have some interesting implications that you could discuss in your introduction and how it relates to your work.

Vulli, A.; et al.. Fine-Tuned DenseNet-169 for Breast Cancer Metastasis Prediction Using FastAI and 1-Cycle Policy. Sensors 2022, 22, 2988.

El-Sappagh, Shaker, et al. "Automatic detection of Alzheimer’s disease progression: An efficient information fusion approach with heterogeneous ensemble classifiers." Neurocomputing (2022).

We have added a description to introduction.

Reviewer 3 Report

Very interesting work, however, several issues require more detail:

1. why the authors did not compare the effects of their work with the commercially available tools for such bioinformatics analyzes 2. whether the developed model will be made available free of charge, and if so, where (website address ) 3.no limitations of the described model are listed, and one of them is, for example, the lack of analysis of the latest publications or the lack of proper validation, especially if it was the application for patient results 5.figure 5 is completely unreadable

Author Response

Response to Reviewer 3 Comments

The authors would like to thank the reviewers and editors for their time and effort to review this manuscript.  Please consider the attached manuscript that has been revised according to the reviews.  Response to each reviewer's comments follows.

  1. why the authors did not compare the effects of their work with the commercially available tools for such bioinformatics analyzes

As far as we know, there are no commercially available tools for XAI to estimate pathogenicity.

I also believe that nothing similar has been published in research.

There are various if only pathogenic estimates that have no explanatory function.

We have added a discussion of these comparisons in section 4.3.

  1. whether the developed model will be made available free of charge, and if so, where (website address )

We plan to apply this methodology to our product and offer it to hospitals. It will not be free, but will be available to the public for a fee.

3.no limitations of the described model are listed, and one of them is, for example, the lack of analysis of the latest publications or the lack of proper validation, especially if it was the application for patient results

We have added explanations to the second half of each of sections 4.4 and 4.5.

5.figure 5 is completely unreadable

To improve the readability of Figure 5, some important nodes have been enlarged.

Round 2

Reviewer 2 Report

.

Author Response

The authors would like to thank the reviewers and editors for their time and effort to review this manuscript.

Reviewer 3 Report

It is very nice that the authors responded to comments in the author's replay in a very succinct manner. The manuscript is prepared quite the opposite. The work is long, confusing, very difficult to follow the described content. It requires re-editing so that the information contained therein is more readable, segregated and presented in a clear and understandable way. Companies such as Roche, ThermoFisher and other bioinformatics companies offer similar tools commercially. Figure 6 must be modified so that the information contained therein can be traced and read. Otherwise, it should be removed, which would be detrimental to the work. Moreover, the description of figures 6 and 7 cannot be reduced to: "Figures 6-7 show an explanation of the decision tree, and Figure 8 shows a human transcription of the decision tree. This explanation is difficult to understand."

Author Response

The authors would like to thank the reviewers and editors for their time and effort to review this manuscript.  Please consider the attached manuscript that has been revised according to the reviews.  Response to each reviewer's comments follows.

It is very nice that the authors responded to comments in the author's replay in a very succinct manner. The manuscript is prepared quite the opposite. The work is long, confusing, very difficult to follow the described content. It requires re-editing so that the information contained therein is more readable, segregated and presented in a clear and understandable way.

We have revised and re-edited 2.3. Evaluation method, 3. Results, 4. Discussion, and 5. Conclusion.

Companies such as Roche, ThermoFisher and other bioinformatics companies offer similar tools commercially.

We have added a discussion of Roche's FoundationOne CDx and ThermoFisher's Oncomine to 4.6.

Figure 6 must be modified so that the information contained therein can be traced and read. Otherwise, it should be removed, which would be detrimental to the work. Moreover, the description of figures 6 and 7 cannot be reduced to: "Figures 6-7 show an explanation of the decision tree, and Figure 8 shows a human transcription of the decision tree. This explanation is difficult to understand."

We have removed Figure 6 and reviewed Figures 7 and 8 (now Figures 6 and 7).

We have reviewed the discussion in 4.2.

Round 3

Reviewer 3 Report

The paper has been revised in accordance with the comments.